# A Single Dose of PEG-Asparaginase at the Beginning of Induction Not Only Accelerates MRD Clearance but Also Improves Long-Term Outcome in Children with B-Lineage ALL

**DOI:** 10.3390/cancers15235547

**Published:** 2023-11-23

**Authors:** Alexander Popov, Günter Henze, Julia Roumiantseva, Oleh Bydanov, Mikhail Belevtsev, Tatiana Verzhbitskaya, Liudmila Movchan, Grigory Tsaur, Svetlana Lagoyko, Liudmila Zharikova, Natalia Myakova, Dmitry Litvinov, Olga Khlebnikova, Olga Streneva, Elena Stolyarova, Natalia Ponomareva, Galina Novichkova, Larisa Fechina, Olga Aleinikova, Alexander Karachunskiy

**Affiliations:** 1Dmitry Rogachev National Medical Research Center of Pediatric Hematology, Oncology and Immunology, 1, S. Mashela St., Moscow 117998, Russia; 2Department of Pediatric Oncology Hematology, Charité—Universitätsmedizin Berlin, 10117 Berlin, Germany; 3Belarussian Research Center for Pediatric Oncology, Hematology and Immunology, 223053 Minsk, Belarus; 4Regional Children’s Hospital, Ekaterinburg 620149, Russia; 5Research Institute of Medical Cell Technologies, Ekaterinburg 620026, Russia; 6Republican Children’s Hospital, Moscow 594268, Russia

**Keywords:** acute lymphoblastic leukemia, PEG-asparaginase, minimal residual disease, flow cytometry

## Abstract

**Simple Summary:**

This report presents the results of the randomized use of PEG-ASP at a dose of 1000 U/m^2^ administered on day 3 of induction therapy in children with B-cell precursor ALL (BCP-ALL) studied for MRD response. At the start of induction therapy, 502 patients were randomized, standard-risk patients into three and intermediate-risk patients into two groups. The single dose of PEG-ASP resulted in a significantly larger proportion of rapidly responding patients. Despite only low or moderately intensive therapy (no or only low-dose anthracyclines, no oxazaphosphorines, no high-dose therapy), these rapid responders also showed excellent long-term outcomes.

**Abstract:**

This report presents the results of the assessment of MRD response by multicolor flow cytometry (MFC) with regard to the randomized use of pegylated asparaginase (PEG). In this study, PEG was randomly administered at a dose of 1000 U/m^2^ on day 3 of induction therapy in children with B-lineage ALL. Methods. Conventional induction therapy consisted of dexamethasone, vincristine, and daunorubicin. MRD data was available in 502 patients who were randomized at the start of induction therapy, standard-risk (SR) patients into three (conventional induction without PEG, induction with additional PEG and with PEG but without daunorubicin) and intermediate-risk (ImR) patients into two groups (with additional PEG and without PEG). Results. The single administration of PEG resulted in a significantly higher proportion of rapid responders, in SR patients even when no anthracyclines were used for induction. In the SR group, the event-free survival of the MFC-MRD fast responders was similar in the PEG− and PEG+ arms (92.0 ± 3.1% vs. 96.2 ± 1.5%, respectively), and the same unfavorable trend was observed for MFC-MRD slow responders (57.5 ± 12.3% vs. 66.7 ± 15.7%, respectively). Results were similar in ImR patients: (94.3 ± 3.2% vs. 95.1 ± 2.4%, for fast responders and 63.3 ± 7.6% vs. 78.1 ± 7.9%, for slow responders in PEG− and PEG+ arms, respectively). However, there is a large difference between the proportion of MFC-MRD slow responders in the PEG− and PEG+ groups (18.3% vs. 5.2% for the SR group and 44.2% vs. 25.0% for the ImR group). Conclusions. Therefore, early use of PEG-ASP not only leads to an accelerated reduction of blasts, but also to an excellent outcome in a significantly larger proportion of patients in both risk groups.

## 1. Introduction

Since the 1960s, asparaginase (ASP) has been used to treat children with acute lymphoblastic leukemia (ALL) [1]. In combination with glucocorticoids, vincristine, and anthracyclines, ASP is a critical component to enhance the effect of induction therapy and improve event-free survival (EFS) [1]. Pegylated ASP (PEG-ASP) has lower immunogenicity and a longer half-life with comparable antileukemic efficacy [2,3]. The main advantages are fewer allergic reactions, fewer administrations due to longer half-life, longer ASP activity, and better overall tolerability. For this reason, PEG-asparaginase is used in many international studies as part of induction therapy [4,5,6,7,8]. In early studies, it was demonstrated by cytomorphology that a single administration of PEG-ASP during the induction phase accelerated the elimination of leukemia cells in the bone marrow (BM) [9]. Response to treatment is now assessed by a much more sensitive measurement of minimal residual disease (MRD) using molecular techniques or multicolor flow cytometry (MFC) [10].

MRD measured by MFC or PCR methods is currently considered one of the most informative and reliable predictors of treatment success [10,11,12,13,14,15,16,17,18]. In almost all modern protocols, MRD assessment at predefined time points (TPs) is used for the most accurate stratification of patients both in conjunction with other prognostically significant parameters and as the sole risk factor [19,20,21,22,23,24,25,26,27,28]. Both types of MRD measurement (cytometric and molecular) have already proven to be very reliable for therapeutic decisions, although each has its own advantages and pitfalls [29,30]. Both MFC-MRD and molecular MRD are fully applicable in multicenter studies [10,29,31,32,33,34,35,36] and form the basis for the development of all innovative treatment approaches [37,38,39]. The type of MRD method used and the most informative TPs depend on the treatment designs and the questions to be answered [12,15,40,41,42].

One aim of the ALL-MB 2008 study was to evaluate the impact of early PEG-ASP administration during induction therapy on long-term EFS. Therefore, patients were randomized into arms that included or did not include early PEG-ASP as intensified induction therapy—three randomization arms for standard-risk (SR) patients and two arms for intermediate-risk (ImR) patients. In parallel, MRD was measured using MFC (MFC-MRD) in a pilot study conducted in clinics affiliated with the laboratories of the Moscow–Berlin Group diagnostic network. This report presents the impact of early PEG-ASP on MFC-MRD outcome measures and, moreover, long-term outcomes in children with B-cell precursor ALL (BCP-ALL).

## 2. Methods

### 2.1. Patients and Risk Groups

The ALL-MB 2008 study is a multicenter study with 50 participating institutions in Russia and Belarus, registered under NCT01953770. This is the third of three consecutive trials for children and adolescents with acute lymphoblastic leukemia (ALL). A total of 3466 unselected patients were registered between February 2008 and November 2014. Of these, 3044 were diagnosed with B-cell precursor ALL (BCP-ALL). Patients were divided into 3 risk groups: standard-risk (SR), intermediate-risk (ImR), and high-risk (HR). The classification criteria had to be easy to collect so that they could be implemented in all participating centers. These were initial white blood cell (WBC) count, splenic enlargement below the costal margin, cytogenetics, and remission status at the end of induction (EOI, day 36).

Standard risk was defined as initial leukocyte count below 30 × 10^9^/L, splenomegaly less than 4 cm below the costal margin, no CNS3 status, no translocation t(4;11)(q21;q23)/*KMT2A::AFF1* or t(9;22)(q34;q11)/*BCR::ABL*, and achievement of hematologic remission at the end of induction [43]. Intermediate-risk patients were those who did not meet SR conditions, did not have any of the specified chromosomal translocations, had an initial leukocyte count of less than 100 × 10^9^/L, and achieved hematologic remission at EOI [44]. All other BCP-ALL patients were assigned to the HR group.

### 2.2. Treatment Protocol

The treatment regimen has been published previously and is shown in Appendix A [43,44,45]. Induction therapy in all patients consisted of vincristine, daunorubicin, dexamethasone, asparaginase, and intrathecal triple therapy. In SR and ImR patients, treatment was continued with three cycles of consolidation therapy followed by maintenance therapy. HR patients, i.e., patients who were not in remission at the time of EOI or initially had a high leukocyte count, received BFM-type HR therapy after induction, followed by BFM protocol II and maintenance therapy [14,45].

### 2.3. PEG-ASP Randomization in Induction

PEG-ASP randomization was only planned for the SR and ImR groups. This randomization schedule is shown in Figure 1. The original induction therapy consisted of vincristine, dexamethasone, and daunorubicin (DNR) plus intrathecal therapy as shown in Figure 1. SR patients were randomly assigned to one of three treatment arms: original induction with DNR without PEG-ASP (PEG − DNR+), PEG-ASP without DNR (PEG + DNR−), and PEG + DNR+. Intermediate-risk patients were randomized to receive either the “classic” ImR induction or the PEG+ arm. In the PEG+ arms, the additional dose of PEG-asparaginase of 1000 U/m^2^ was administered on day 3. After achieving complete remission at the end of induction (EOI, day 36), children received three cycles of consolidation followed by maintenance therapy as previously described [46].

### 2.4. MRD Investigation

For logistical reasons, measurement of MRD in patients from all participating institutions in Russia and Belarus was unrealistic in the ALL-MB 2008 study. Therefore, the MFC-MRD pilot study presented here was performed with patients eligible for treatment only in facilities affiliated with the MFC laboratories of the Moscow–Berlin group flow network [16]. Bone marrow samples for MFC-MRD monitoring were collected on day 15 and/or EOI. MRD was measured by MFC in three laboratories (two in Russia and one in Belarus) using the well-harmonized approach based on AIEOP-BFM-ALL-MRD-Flow study group guidelines [47], as previously described [48]. All three laboratories use the MFC methodology based on standard analyses, and had participated in AIEOP-BFM QA system [34] as well as in intragroup proficiency tests [48]. A 4–9-color MFC was used to evaluate the expression of antigens commonly used for MRD detection in BCP-ALL: CD19, CD10, CD34, CD45, CD20, CD38, CD58, and CD11a [48]. EuroFlow guidelines for monitoring machine performance were used [49]. At least 300,000 nucleated cells were examined. Lymphoblasts were classified as leukemic if they represented a distinct population with leukemia-associated phenotypes and lymphoid light scattering parameters. Following the I-BFM-FLOW network guidelines [34] and in accordance with the standardized EuroFlow approach [33] [20], we defined a minimum of ten clustered leukemic lymphoblasts to consider the cell population as leukemic. The MRD values were expressed as percentage of leukemia cells among all nucleated bone marrow cells which were defined by positivity for nucleic acid staining (Syto16 or Syto41 dye). MRD negativity was defined as <0.01%. Despite the increasing number of colors in use, the basic principles of MFC-MRD detection had not changed over time. This sensitivity was thus achievable with high reliability over the entire study period.

### 2.5. Statistical Analysis

Pearson’s chi-square test was used for qualitative comparisons. The event-free survival (EFS) was defined as the time from diagnosis to the first event or to the last contact if no events were reported. Failure to respond, relapse, death from any cause, or a second malignant neoplasm were considered adverse events. EFS curves with the Greenwood standard errors [50] were constructed using the Kaplan–Meier method [51]. The log-rank test was used to compare the results between groups. Cumulative incidence of recurrence (CIR) curves were calculated taking into account the competing risk for other relevant events [52] and compared with Gray’s test [53]. All tests were two sided. The *p*-value < 0.05 was considered significant. Analyses were performed using R-statistics v3.4.2.

## 3. Results

### 3.1. Overall Description

A total of 1702 children with BCP-ALL were stratified into the SR group and 1105 into the ImR group. Of these, MFC-MRD was investigated in 295 and 227 patients, respectively, at least one in time-point (TP). The final distribution of patients with available MFC-MRD data in relation to randomization at induction is shown in the CONSORT diagram (Figure 2). A total of 504 were examined on day 15 and 493 at EOI. Of the children evaluated for MFC-MRD, 288 of 295 (97.6%) SR patients and 214 of 227 (94.3%) ImR patients were randomized. The characteristics of these patients are presented in Table 1.

### 3.2. MRD Elimination in the Randomization Arms

The distributions of the MFC-MRD values in the randomization arms for both TPs are shown in Figure 3. For both TPs, the MFC-MRD values were categorized qualitatively (positive/negative) and quantitatively according to previously defined threshold values with the strongest influence on the prognosis [44].

In both risk groups, the rate of MFC-MRD elimination is significantly faster in the PEG-ASP arms. In the SR group, MFC-MRD was significantly more often positive at day 15 in the DNR+/PEG− randomization arm (Figure 3A; *p* < 0.001). The same is true when comparing the incidence of high (greater than 1%) MFC-MRD positivity at day 15 (*p* < 0.001). Even in the arm without anthracyclines (PEG+/DNR−), MFC-MRD elimination was faster than in the PEG−/DNR+ arm (*p* < 0.001 and *p* = 0.001, respectively). Although four-drug induction resulted in slightly faster MFC-MRD elimination, the differences between the PEG+/DNR− and PEG+/DNR+ groups were not statistically significant: *p* = 0.251 for qualitative and *p* = 0.076 for quantitative comparisons.

The EOI-MFC-MRD results (Figure 3B) also differed between the PEG+ and PEG− arms. MFC-MRD rates were higher in the DNR+/PEG− arm (*p* < 0.001) and the proportion of patients in the MFC-MRD group with slow responders (over 0.1%) was also higher (*p* = 0.003). Again, no differences were found between the two PEG+ arms (*p* = 0.478 and *p* = 0.590), and again both MFC-MRD positivity and MFC-MRD slow response in DNR+/PEG− were higher than in the PEG+/DNR− arm (*p* = 0.003 and *p* = 0.017, respectively).

In the ImR group, the single administration of PEG-ASP also accelerated the clearance of MFC-MRD. At day 15 (Figure 3C), MFC-MRD positivity was more common in the PEG− arm (*p* < 0.001), as was the proportion of patients with slow MFC-MRD response (MFC-MRD ≥ 1%) (*p* < 0.001). For EOI, the results (Figure 3D) were comparable: MFC-MRD positivity was higher in the PEG− group (*p* = 0.004), and the most meaningful threshold for these ImR patients was MFC-MRD negativity (0.01%) [44].

### 3.3. Prognostic Value of MRD Data in Different Arms

Results from the 2008 ALL-MB study have already shown that EOI-MFC-MRD data have crucial prognostic value and potential clinical relevance [44]. Nevertheless, the early blast cell reduction rate as evidenced by MFC-MRD values at day 15 also showed significant prognostic value with the most informative threshold of 1% [43,44]. Therefore, we focused on both TPs to analyze the relationship between the predictive value of MFC-MRD and PEG-ASP randomization during induction. In the SR group, the two PEG+ arms were analyzed together due to the similar MFC-MRD clearance (see Figure 3). EFS and CIR curves for MFC-MRD results at day 15 by randomization arm are shown in Figure 4. In the SR group, the proportion of fast and slow MFC-MRD responders in the PEG− arm was almost equal, and their outcome was not significantly different (Figure 4A). In contrast, in the PEG+ arms (Figure 4B), the difference in outcome between the majority of patients diagnosed with low MFC-MRD (<1%) and the few patients with slow MFC-MRD response was much more pronounced. In the ImR group, the situation was reversed (Figure 4C,D). In the PEG+ arm, the outcome was relatively good even for children with a slow early MFC-MRD response (day 15 MFC-MRD ≥ 1%), whereas in the PEG− arm the difference between fast and slow MFC-MRD responders was quite marked.

EFS and CIR curves for the EOI-MFC-MRD results by randomization arm are shown in Figure 5. In the SR group, the outcome of the MFC-MRD fast responders in the PEG− (Figure 5A) and PEG+ arms (Figure 5B) was comparable, and the trend for the MFC-MRD slow responders was equally unfavorable. However, there is a large difference between the proportion of MFC-MRD slow responders in these two groups (see Figure 3). The situation is similar in the ImR group (Figure 5C,D). In patients in both risk groups, the early use of PEG-ASP leads to an accelerated decline in blast cells and an excellent overall outcome. Still, the proportion of patients with favorable MRD values is higher in the PEG+ arms. In the ImR group, the superior EFS and CIR particularly impacted patients in the PEG+ arm, whereas the difference between the arms in the SR group was not as obvious (Appendix A).

### 3.4. PEG-ASP Related Toxicity

Toxicity was comparable between the treatment arms; with the exception of reduced fibrinogen levels in the SR groups, there were no significant differences (Table 2).

## 4. Discussion

ASP was first described by Oettgen et al. in 1967 [54], and is therefore one of the oldest drugs used as an essential component of induction therapy for childhood ALL. ASP is not a cytostatic drug, but still has a number of undesirable side effects. Most common are hypersensitivity reactions, which sometimes force premature discontinuation of therapy. Other serious side effects include clotting disorders and (necrotizing) pancreatitis. Toxicity to the pancreas includes interactions of ASP with insulin and may lead to insulin-dependent hyperglycemia. Therefore, efforts were made to avoid concomitant administration of ASP and glucocorticoids during induction therapy and to postpone the use of ASP. This principle was also applied in the first Russian multicenter ALL study “ALL-MB 91” [55]. The results of a first randomized trial in Russia comparing native ASP at a dose of 5000 versus 10,000 U/m^2^ for standard-risk patients during consolidation therapy were published in 2019 [56]. No advantage for the higher dose could be demonstrated. The 5000 U dose was found to be equally effective, less expensive, and less toxic than 10,000 U/m^2^ for SR patients.

A pegylated ASP product was developed to prevent or reduce hypersensitivity and thus minimize the consequences of treatment discontinuation in a large number of patients. In fact, PEG-ASP fundamentally changed therapy in this regard. PEG-ASP has lower immunogenicity and a longer half-life with comparable antileukemic efficacy [1]. Therefore, its use has been recommended by many international groups for frontline treatment [4,5,6,7,8,40]. PEG-ASP is now the most commonly used asparaginase product in modern ALL protocols.

In early studies, cytomorphological data showed that a single dose of PEG-ASP in the induction phase markedly accelerated the elimination of blasts in the bone marrow of children with BCP-ALL [9]. The present study confirms this by measuring the decline in leukemia cells using the much more sensitive MFC-MRD technique. The results published by Avramis et al. are quite comparable to our data on day 15, as day 7 of the 1952 and 1962 CCG protocols more or less corresponds to day 15 of the 2008 ALL-MB protocol [9]. Furthermore, our threshold for discriminating between fast and slow responders in this TP is 1%, which is the same order of magnitude value as the definition of M1 BM status with <5% blasts. Thus, we show that even the relatively moderate dose of 1000 U/m^2^ PEG-ASP used here results in as rapid an early response as a 2.5-fold higher dose as recommended and used later.

Furthermore, the rapidity of early response in the SR group was the same even among children who had not received anthracyclines at all during the induction phase. The MFC-MRD response on day 15 retained its prognostic value only in the PEG+ arms of the SR group and the PEG− arms of the ImR group. This finding only confirms the limited applicability of MFC-MRD data at day 15 in our protocol, as previously shown [43,44]. If indeed no prognostic significance of MFC-MRD was detectable in either situation, nevertheless the outcome in the “slow” responders was not so poor as to be considered prognostically unfavorable. Still, in most pediatric ALL protocols, the most clinically relevant TP for MFC-MRD measurement is the EOI [11,13,18,19,33,57]. It discriminates well between fast and slow MFC-MRD responders and excludes very high-risk patients who did not achieve remission at EOI. For the ALL-MB 2008 trial, this TP also has the greatest clinical relevance [46], so we focus mainly on the results obtained at this TP. At the EOI, the rapid MFC-MRD response with previously defined thresholds (different for SR and ImR groups [46]) retained its ability to discriminate patients for whom low- and moderate-intensity treatment is sufficiently effective. Thus, the importance of early administration of PEG-ASP is not limited to the increased rate of MFC-MRD elimination, but also affects the subsequent treatment strategy.

The faster elimination of MFC-MRD can easily be explained by the addition of another drug to the induction regimen. However, the effect of PEG-ASP is obviously not limited to the faster MFC-MRD response; the long-term outcome in the fast responders is still significantly better in the PEG+ arms than in the slow responders of the same arms, and in addition, the higher proportion of MFC-MRD fast responders after PEG+ also leads to an improvement in overall outcomes. Here, PEG-ASP was administered at a moderate dose of only 1000 U/m^2^. Later, other protocols recommended a dose of 2500 U/m^2^ based on more extensive pharmacologic data [58]. However, even the moderate dose used here during induction results in a significantly higher proportion of fast responders with excellent long-term outcome, despite overall low (SR) or moderate (ImR) treatment intensity. As a result, significantly fewer patients need to be switched to more intensive therapies due to a slower response [44] and the overall results improve due to the larger group with a lower relapse rate. In the ImR group this improvement is evident, while in the SR group the trend is not so clear.

Notably, in the SR group, in the anthracycline-free PEG+ arm, the proportion of MFC-MRD fast responders was high and the EFS was excellent. Only 7.0% of the children in this group received a single dose of DNR on day 22 due to a slow response to treatment by day 15 (compared to 16.5% in the PEG− group and 6.1% in the four-drug group). Thus, in a large proportion of patients, anthracyclines, one of the most toxic drugs against leukemia, can be dispensed with without reducing the survival rate. This finding is consistent with recently published COG data showing remarkable success of anthracycline-free induction therapy in low-risk (LR) patients [8,58,59]. In the AALL0932 trial, children who met COG-LR criteria were treated with three-agent induction with dexamethasone, vincristine, and early PEG-ASP at a dose of 2500 U/m^2^ [58]. The achieved 5-year disease-free survival of 98% confirms that for children with LR BCP-ALL, this three-agent induction is the adequate therapy, starting with mandatory control at EOI. However, the ALL-MB 2008 SR criteria differ from the COG LR criteria [43]. Although we use slightly more stringent baseline clinical parameters, we propose a higher threshold for the EOI MFC-MRD (0.1% instead of 0.01%) and thus do not isolate an identically sized group of patients with rapid response and favorable clinical presentation. As shown previously, we can identify up to 50% of all BCP-ALL patients with an excellent outcome, despite only low-intensity therapy [43]. In addition, in both the MFC-MRD trial and the overall randomized trial (manuscript in preparation), we did not find an advantage of four-drug induction compared with a three-drug PEG+ regimen in this cohort.

In contrast to the SR group, in the ImR group of the ALL-MB 2008 trial, accelerated clearance of MFC-MRD in the PEG+ arm directly improved EFS. This difference, which was clearly seen in the limited cohort of patients screened for MFC-MRD, it remained in the full cohort study (manuscript in preparation). Since ImR is always one of the most debatable groups in all ALL trials [60], we think that accelerating MFC-MRD elimination after early use of a moderate dose of PEG-ASP is a promising way to improve outcomes in ImR patients as well.

## 5. Conclusions

In conclusion, the addition of a single dose of as little as 1000 U/m^2^ of PEG-ASP at the start of induction significantly accelerates MFC-MRD clearance, resulting in excellent EFS in rapidly responding patients and an improvement in overall outcomes in children with BCP-ALL when treated with an overall low or moderate intensity protocol.

## Figures and Tables

**Figure 1 cancers-15-05547-f001:**
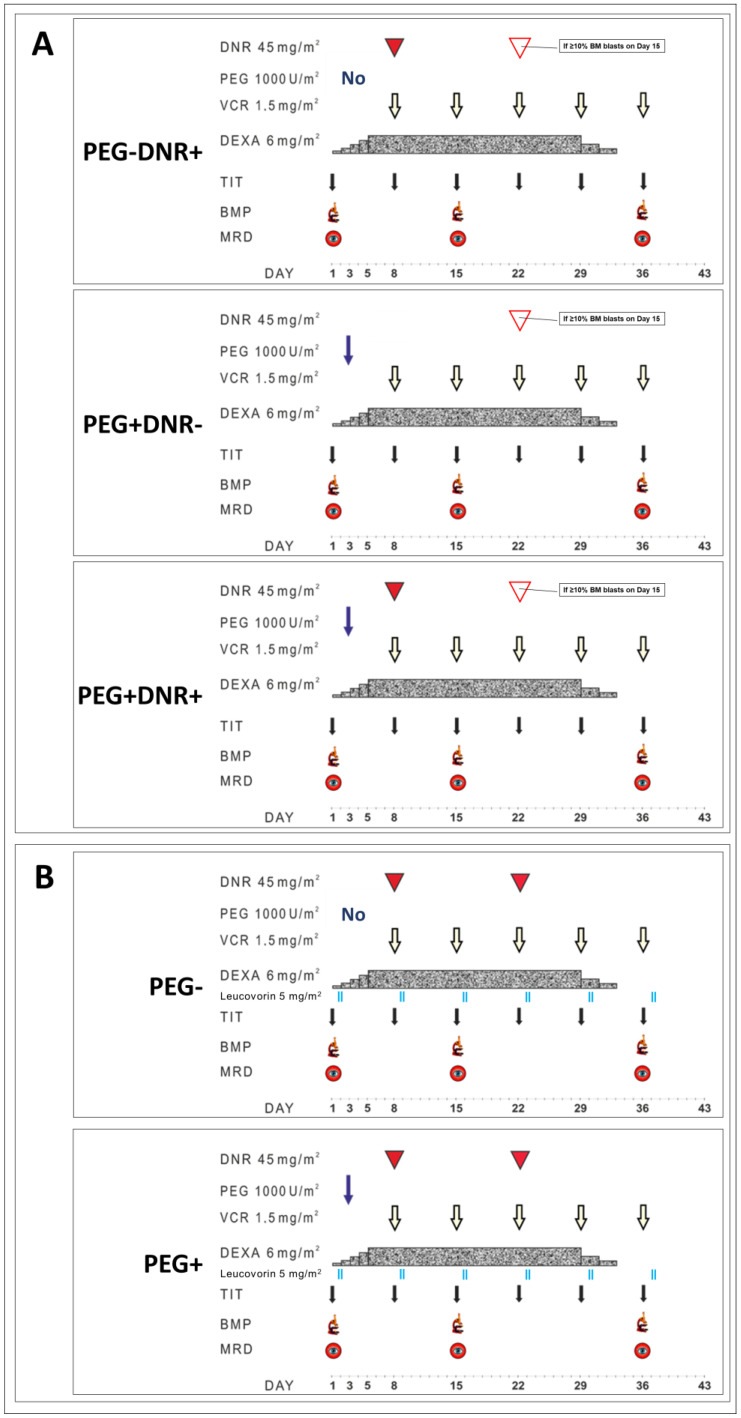
Details of the induction therapy with randomization arms in SR (**A**) and ImR (**B**) groups.

**Figure 2 cancers-15-05547-f002:**
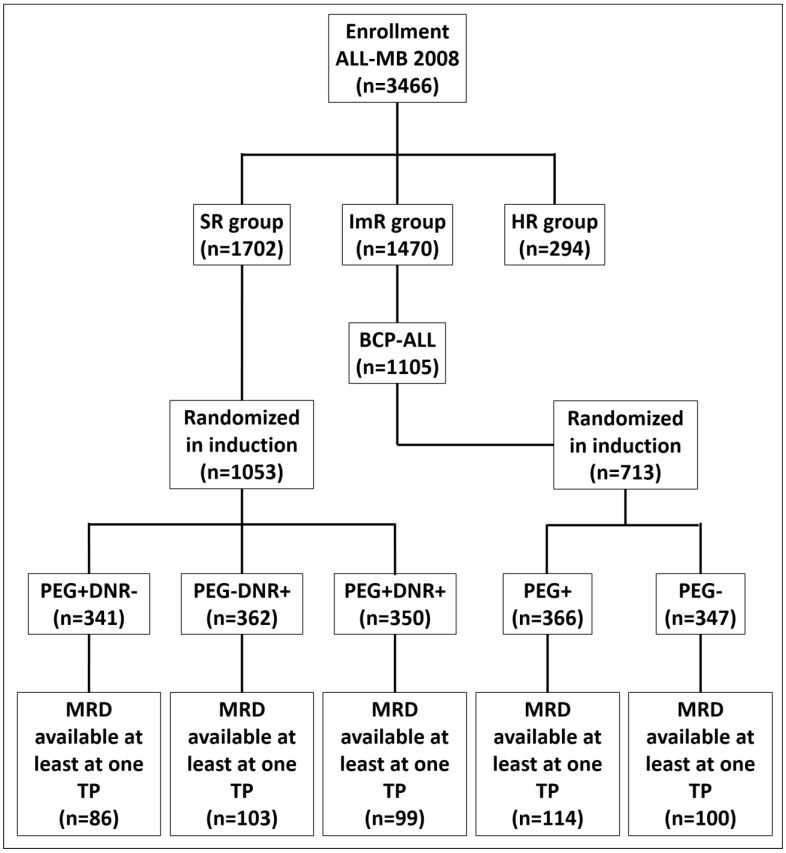
CONSORT diagram.

**Figure 3 cancers-15-05547-f003:**
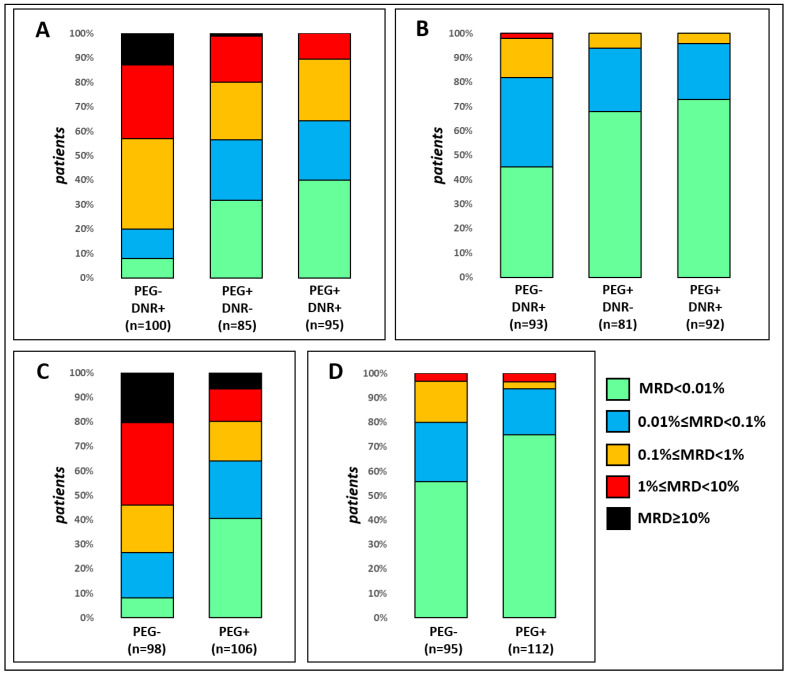
Distribution of MFC-MRD levels in different randomization arms in the SR group (**A**,**B**) and in the ImR group (**C**,**D**). Day 15 data is displayed in panels (**A**,**C**), while EOI data is displayed in panels (**B**,**D**).

**Figure 4 cancers-15-05547-f004:**
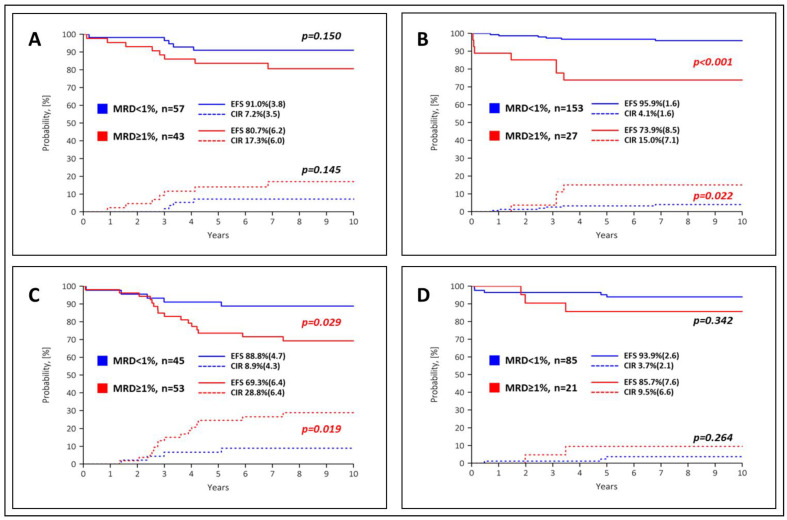
Event-free survival (EFS, solid lines) and cumulative recurrence incidence (CIR, dashed lines) according to MFC-MRD values for day 15 in the SR group (**A**,**B**) and in the ImR group (**C**,**D**) with the 1% threshold. Images (**A**,**C**) show PEG− arms, images (**B**,**D**) show PEG+ arms. Standard errors are given in parentheses.

**Figure 5 cancers-15-05547-f005:**
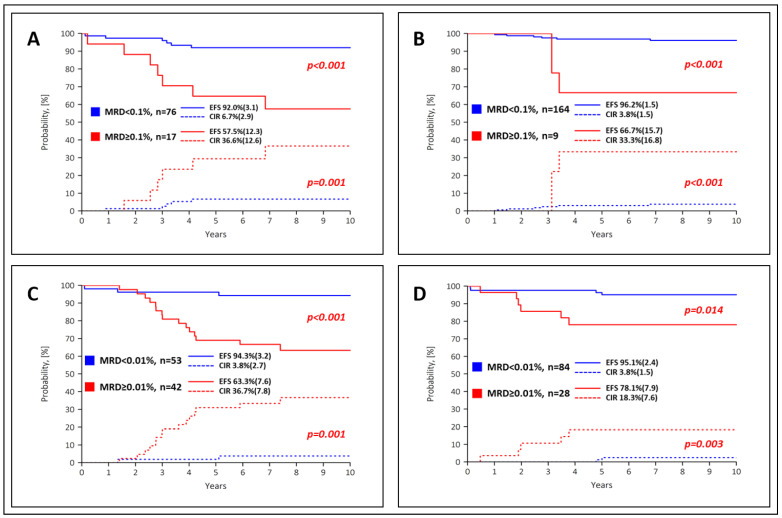
Event-free survival (EFS, solid lines) and cumulative recurrence incidence (CIR, dashed lines) according to MFC-MRD values for EOI in the SR group (**A**,**B**) with the 0.1% threshold and in the ImR group with the threshold of 0.01% (**C**,**D**). Images (**A**,**C**) show PEG− arms, images (**B**,**D**) show PEG+ arms. Standard errors are given in parentheses.

**Table 1 cancers-15-05547-t001:** Characteristics of patients enrolled in the MRD study who were randomized to receive PEG-asparaginase (PEG) on day 3 during induction therapy. Panel A shows patients from the SR group (n = 288), panel B shows the ImR group (n = 214). Bolds are used to indicate significant values.

Panel A	PEG + DNR−		PEG + DNR+		PEG − DNR+		*p* *
	n	%	n	%	n	%	
Total	86	100	99	100	103	100	n/a
Sex							
Male	45	52.3	49	49.5	55	53.4	0.850
Female	41	47.7	50	50.5	48	46.6	
Age							
<10 y.o.	73	84.9	83	83.8	87	84.5	0.981
≥10 y.o.	13	15.1	16	16.2	16	15.5	
Glucocorticoid response **							
Good	85	98.8	97	98.0	98	95.2	0.333 ***
Poor	0	0.0	0	0.0	2	1.9	
ND	1	1.2	2	2.0	3	2.9	
Day 15 bone marrow response (by cytomorphology) ^#^							
M1	73	84.9	83	83.8	71	68.9	**0.012**
M2	11	12.8	14	14.2	22	21.4	
M3	2	2.3	1	1.0	9	8.7	
ND	0	0	1	1.0	1	1.0	
t(12; 21) (p13; q22)/*ETV6::RUNX1*							
Present	22	25.5	25	25.3	33	32.0	0.396
Absent	63	73.3	73	73.7	66	64.1	
ND	1	1.2	1	1.0	4	3.9	
**Panel B**	PEG+			PEG−			*p* *
	n		%	n		%	
Total	114		100	100		100	n/a
Sex							
Male	56		49.1	51		51.0	0.784
Female	58		50.9	49		49.0	
Age							
<10 y.o.	92		80.7	86		86.0	0.301
≥10 y.o.	22		19.3	14		14.0	
Initial WBC count							
<50 × 10^9^/L	90		78.9	78		78.0	0.866
≥50 × 10^9^/L	24		21.1	22		22.0	
Glucocorticoid response **							
Good	108		94.8	93		93.0	0.571
Poor	3		2.6	4		4.0	
ND	3		2.6	3		3.0	
Day 15 bone marrow response (by cytomorphology) ^#^							
M1	98		86.0	61		61.0	**<0.001**
M2	8		7.0	20		20.0	
M3	8		7.0	17		17.0	
ND	0		0.0	2		2.0	
t(12; 21) (p13; q22)/*ETV6::RUNX1*							
Present	17		14.9	16		16.0	0.749
Absent	96		84.2	80		80.0	
ND	1		0.9	4		4.0	
CNS leukemia (CNS3-status)							
Present	4		3.5	5		5.0	0.588
Absent	110		96.5	95		95.0	

* Patient distributions were compared with two-sided chi-square test. ** Poor glucocorticoid response: blast count in peripheral blood ≥ 1000 cells per µL on day 8. *** Patient distribution was compared with two-sided exact Fisher’s test. ^#^ M1 bone marrow status was defined as leukemia cells < 5%; M2—leukemia cells 5–25%; M3—leukemia cells ≥ 25%. ND—no data, DNR—daunorubicin.

**Table 2 cancers-15-05547-t002:** Incidence of adverse effects in the induction phase in 492 patients studied for whom toxicity data were available at this stage of treatment. Panel A shows the data for the SR group; Panel B for the ImR group. Bolds are used to indicate significant values.

Panel A	PEG − DNR+		PEG + DNR+		PEG + DNR−		*p*
	n	%	n	%	n	%	
Total number of patients	101	100	99	100	83	100	
Amylase (>2–5 × UNL)	0	0	2	2.0	2	2.4	0.317
Fibrinogen (>40% reduction)	3	3.0	13	13.1	11	13.3	**0.020**
Thrombosis (deep vein thrombosis, need AC therapy or embolism)	2	2.0	1	1.0	1	1.2	0.829
Hyperglycemia (>13.9 mmole/L)	1	1.0	2	2.0	1	1.2	0.812
Infection	58	57.4	57	57.6	52	62.6	0.725
**Panel B**	PEG−			PEG+			*p*
	n	%		n	%		
Total number of patients	111	100		98	100		
Amylase (>2–5 × UNL)	1	0.9		3	3.1		0.255
Fibrinogen (>40% reduction)	10	9.0		12	12.2		0.447
Thrombosis (deep vein thrombosis, need AC therapy or embolism)	1	0.9		1	1.0		0.929
Hyperglycemia (>13.9 mmole/L)	3	2.7		7	7.1		0.133
Infection	69	62.2		66	67.3		0.434

UNL—upper normal limit; AC—anticoagulant.

## Data Availability

The datasets generated during and/or analyzed during the current study are available from the corresponding author on reasonable request.

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
