# Peer review of "A Single Dose of PEG-Asparaginase at the Beginning of Induction Not Only Accelerates MRD Clearance but Also Improves Long-Term Outcome in Children with B-Lineage ALL"

_cancers, 2023, doi:10.3390/cancers15235547_

Round 1

Reviewer 1 Report

Comments and Suggestions for Authors

This is a study without major flaws.

The data is historic, of moderate interest and originality, still seems worth publishing again confirming the significant activity of asparaginase. On the other hand, treatment, stratification and monitoring of ALL was much improved in the meantime.

As the data are mature, and the authors discuss toxicities as a major problem with asparaginase treatments, there should be a section on toxicity in the reporting. Especially asparaginase-specific: thrombosis, pancreatitis, dyslipidemia, hyperbilirubinemia, future allergic reactions. I must have been collected in the trial.

Comments on the Quality of English Language

English is fine.

Author Response

Response to Referee 1

We thank the reviewer for the valuable comment. We have added the table (Table 2) with the available data regarding induction toxicity in the study group. As all three participated institutions belonged to the “core” group of the ALL-MB 2008 study, the respective data is available for nearly all patients.

Reviewer 2 Report

Comments and Suggestions for Authors

The Authors report the positive impact of PEG-asparaginase early administration in the treatment of pediatric ALL on MRD clearance and outcome. This is an interesting result also considering the lower PEG-ASP dosage as compared to other international protocols.

I believe that the paper might be improved as regard to clarity of presentation of results.

Following find my comments:

Abstract: line 32 the sentence is not clear, should be better explained. Line 33: the authors cite qualitative and quantitative comparison (MRD?) however this distinction is not discussed during the paper, therefore it should be avoided.

Introduction: A brief paragraph should be added explaining the importance of MRD, since this is a key point of the paper. 

Results: figure 3, legend: the signs indicating MRD "lower of equal to"  should be rotated, since they should indicate "greater or equal". This applies to the blue, yellow and red box.

Paragraph 3.3: line 201 and 220: the expression "see above" should be substituted with "see figure 3" for example 

Figure 4: the legend to the table states that the table refers to "MRD EOI", however in the text the authors state "MRD +15"

Line 221-223: this sentence is not clear

Discussion: line 266:   "The MFC-MRD response at day 15 retained its prognostic value only in the PEG+ arms of the SR group"  probably because MRD slow responders in this arm are patients who need intensification of treatment soon after day 15. Please discuss.

Line 279: this statement is not clear, I mean I could not find such result in the paper

Comments on the Quality of English Language

Dear Editor, thank you for giving me the opportunity of reading this paper.

This is an interesting paper, reporting important results to the treatment of pediatric ALL, however it could be improved 

Author Response

Response to Referee 2

Response to the reviewer

We thank the reviewer for their valuable comments and are sure that they will improve the text.

Comment 1.

Abstract: line 32 the sentence is not clear, should be better explained. Line 33: the authors cite qualitative and quantitative comparison (MRD?) however this distinction is not discussed during the paper, therefore it should be avoided.

Response

We have changed and corrected the text in the abstract as suggested by the reviewer.

Comment 2.

Introduction: A brief paragraph should be added explaining the importance of MRD, since this is a key point of the paper.

Response

We have added the requested paragraph as recommended (lines 64-74)

Comment 3.

Results: figure 3, legend: the signs indicating MRD "lower of equal to"  should be rotated, since they should indicate "greater or equal". This applies to the blue, yellow and red box.

Response

This legend shows that we have used five different ranges of MFC-MRD values. The three descriptions above mean that this group contains patients with MFC-MRD values that are greater than or equal to the lower value (e.g., 0.01% for the "blue" group) and significantly less than the upper value (e.g., 0.1% for the "blue" group). According to this definition, we consider this type of presentation to be the most suitable for the legend, which is located directly on the figure.

Comment 4.

Paragraph 3.3: line 201 and 220: the expression "see above" should be substituted with "see figure 3" for example

Response

Amended as proposed.

Comment 5.

Figure 4: the legend to the table states that the table refers to "MRD EOI", however in the text the authors state "MRD +15"

Response. Corrected.

Comment 6.

Line 221-223: this sentence is not clear

Response.

We have reworded this sentence in the revised text (lines 234-237).

Comment 7.

Discussion: line 266:   "The MFC-MRD response at day 15 retained its prognostic value only in the PEG+ arms of the SR group"  probably because MRD slow responders in this arm are patients who need intensification of treatment soon after day 15. Please discuss.

Response.

We have added our thoughts and assessment of this statement in lines 289-293.

Comment 8.

Line 279: this statement is not clear, I mean I could not find such result in the paper

Response.

The wording was unclear, and the message of this statement was misleading. We have reworded this sentence to make the message clear and correct (lines 304-308).

Round 2

Reviewer 1 Report

Comments and Suggestions for Authors

The manuscript is satisfactory now.

Comments on the Quality of English Language

No major problems.